# Grazing Cattle, Sheep, and Goats Are Important Parts of a Sustainable Agricultural Future

**DOI:** 10.3390/ani12162092

**Published:** 2022-08-16

**Authors:** Temple Grandin

**Affiliations:** Department of Animal Science, Colorado State University, Fort Collins, CO 80526, USA; cheryl.miller@colostate.edu; Tel.: +1-970-310-5411

**Keywords:** sustainability, soil health, rotational grazing, cover crops, livestock

## Abstract

**Simple Summary:**

Increasing attacks on animal agriculture have forced many people to question the use of animals for food. Grazing livestock are part of a sustainable agricultural future. Vast amounts of land all around the world can only be used for grazing. It is either too arid or the terrain is too rough for growing crops. Rotating cattle, sheep, or other livestock between different pastures can improve both soil health and plant biodiversity. This is a sustainable use of land that cannot be cropped. On cropland, the rotation of conventional crops, such as corn or soybeans, with livestock grazing on a forage crop can improve both soil health and reduce the need for artificial fertilizer. Successful grazing programs must be adapted to local conditions. When grazing is performed correctly, it will improve the land.

**Abstract:**

Many people believe that animal agriculture should be phased out and replaced with vegetarian substitutes. The livestock industry has also been attacked because it uses vast amounts of land. People forget that grazing cattle or sheep can be raised on land that is either too arid or too rough for raising crops. At least 20% of the habitable land on Earth is not suitable for crops. Rotational grazing systems can be used to improve both soil health and vegetation diversity on arid land. Grazing livestock are also being successfully used to graze cover crops on prime farmland. Soil health is improved when grazing on a cover crop is rotated with conventional cash crops, such as corn or soybeans. It also reduces the need for buying fertilizer. Grazing animals, such as cattle, sheep, goats, or bison, should be used as part of a sustainable system that will improve the land, help sequester carbon, and reduce animal welfare issues.

## 1. Introduction

This paper contains a combination of observations I have made during a fifty-year career in the cattle industry, combined with references from scientific research. I have visited ranching operations in many different countries during my work as a livestock handling consultant. Recently, I taught a seminar to a group of animal science students at the University of Nebraska. We entered into an intense discussion about the future sustainability of animal agriculture. I told the students that I have been in the cattle industry for 50 years. I have worked hard to improve livestock handling, design equipment, and implement animal welfare auditing programs [1,2]. My work in animal agriculture has improved livestock handling. Unfortunately, it has been attacked by some people in the animal rights movement. They believe that using animals for food should be abolished. The animal advocacy movement has shifted toward veganism [3]. Another issue that concerns many people is that scientists are also learning that cattle, pigs, octopuses, and other animals are sentient beings [4,5]. This knowledge has also increased ethical concerns about how people use animals. I have always believed that cattle and other animals are conscious. On many ranches, I have observed good stock people having positive interactions with their cattle. At ranches in Texas and Australia, I have watched Brahman cows approach people to be petted and scratched. The animal was motivated to initiate human contact and not food. During a long career, I have observed increasing societal concerns on how animals we use for food are treated [6]. Many people currently believe that animal agriculture should be phased out and replaced with vegetarian substitutes or meat cells grown in bioreactors. There are now many scientific papers about cultured meat [7,8,9].

Another concern is the effect of livestock on the environment. Cattle definitely emit methane, but there are other sources such as wetlands, leaking oil field equipment, and landfills [10]. Wetlands may emit one third of atmospheric emissions of methane [11]. Before Europeans came to North America, large herds of bison emitted large amounts of methane. Bison methane emissions in the past would have been equal to 86% of farmed ruminants in the United States [12].

I have visited ranches and observed sheep and cattle grazing in every state and province in the U.S. and Canada. In Central and South America, I have been to Mexico, Brazil, Argentina, Uruguay, and Chile. In Europe, I have visited cattle pasture operations in the United Kingdom, Ireland, France, Denmark, and Portugal. Other places I have visited are the Australian Outback and the green hills of New Zealand. I have learned from my extensive travels that there are vast amounts of land that can only be used for grazing. The first type of land is too arid and hilly to raise crops. Some examples are the high desert in Arizona, Sand Hills in Nebraska, Texas Panhandle, and the Australian Outback. The second type of grazing land has higher moisture but it is also too hilly and rough for growing crops. Some examples in the U.S. would be the hill country in eastern Kentucky and southern Missouri. The green, steep hills of New Zealand are another place where grazing is the only option. During my travels, I have seen the very best and the very worst grazing operations. I have talked to many family ranchers who care deeply about the land. They are proud of being good stewards of both the land and their animals.

All the issues that have just been discussed have motivated me to think deeply about the use of animals for food. Ruminant animals are the only way to produce food on these lands. On a well-managed grazing operation, cattle and sheep welfare would be better compared to either indoor confinement or a muddy feedlot. When I started learning more about plants and agronomy, I discovered that grazing is part of the natural grassland system. My conversation with the students shifted to their own state of Nebraska. Nebraska is an excellent starting point for discussion due to its diverse land types. The western sandhills are too dry and hilly to be cropped. The eastern part of the state is prime farmland that is being used to grow cash crops, such as corn (maize) and soy.

## 2. Insight from Learning That Animals Are Part of the Land

A few years ago, I attended a departmental seminar at my university. A visiting agronomist explained how grazing bison created the best U.S. cropland in Iowa and Illinois. This was an important insight for me and I learned about how grazing animals were a natural part of the land. The next section of this paper discusses scientific literature that supports the concept of the use of grazing to improve the land. Richard Teague from Texas A&M University explains that grasslands evolved with grazing animals. There were intense periods of grazing followed by a long period of rest, which allowed the pasture fully recover [13]. There is evidence that grazing can also be used to improve soil health and sequester carbon [13,14,15,16]. Another sustainability benefit of well-managed grazing is increased plant biodiversity [15,16,17]. There is also an abundance of scientific evidence that shows that rotating crops, such as corn or soy, with livestock grazed on cover crops can improve soil health [18], sequester carbon [14,19], and increase the abundance of pollinating insects [20]. High-intensity agriculture with low crop diversity will decrease insect biodiversity [21]. A long-term study conducted in Ontario Canada showed that soybean monocultures were detrimental to soil health [22].

I told the students that they should work on developing systems to integrate grazing animals, such as cattle, sheep, goats, or bison, into both pasture- and crop-based systems. Nebraska is a unique state that contains both prime cropland in the eastern half for non-irrigated dryland cash-crop farming and western sand-hills land. The sand hills are the largest field of dunes in the Western Hemisphere [23]. It is stabilized by surface vegetation that can be grazed by livestock. They are unique land that is not suitable for crops. It is too hilly and underground water supplies are limited. At present, the sand hills are used for grazing livestock. There are also many other parts of the world where the land is either too arid or lacks a sufficient underground water supply to raise crops [24]. Estimates show that at least 20% of the habitable land on Earth would only be suitable for grazing [25]. I informed the students that we needed to continue to grow food on the sand hills with grazing livestock. They also needed to work with the farmers on the prime farmland in eastern Nebraska to integrate grazed cover crops with standard cash crops, such as corn (maize) or soy.

## 3. Basic Principles of Rotational Grazing

There are some basic principles of grazing, but I always tell producers to obtain good local advice. A method that works well and benefits the soil in one part of the world may not work in another. Grazing in different regions often needs to be managed differently [26]. The basic principle of intense rotational grazing is to mimic the way herds of wild herbivores, such as bison or wildebeest, grazed the land [27,28]. A herd of ruminants would migrate in a tight bunch. The animals grazed a patch of land and then moved on. They would “mow” the vegetation and then not return until next year. This would provide an entire year for the plants to regenerate. One of the early rotational grazing mistakes was not giving the pasture sufficient time to recover until it was grazed again [29,30]. Modern electric fencing that can run on solar power has made it much easier and more economical to create small paddocks. The animals can be bunched relatively densely to force them to eat all types of vegetation before being moved. This prevents the problem of animals eating only the highly palatable plants and leaving the less palatable plants behind. Grazing specialist Fred Provenza calls this “eat the best and leave the rest” [31]. There is a basic principle of plant physiology that many people forget. When a pasture regrows, the green leaves recover before the roots [29,30]. The recovery time has to be sufficiently long to allow the roots to fully recover.

### 3.1. Four Basic Types of Grazing

In the scientific literature, there are many different terms being used to describe grazing systems. The four basic types are (1) continuous grazing on a single pasture; (2) conventional rotation between two to four pastures, which is the most common system in low-rainfall areas; (3) multiple paddocks where the animals are moved every four to eleven days; and (4) mob grazing, also called adaptive multi paddock [13,16]. In mob grazing, the animals are stocked very densely and they may need to be moved either once or twice a day. In many parts of the U.S. with higher rainfall levels, the most common system is using paddocks where the livestock are moved every few days. This is especially true in the more wetter parts of the U.S.

### 3.2. Soil Health Benefits of Rotational Grazing

The full soil health benefits of rotational grazing cannot be achieved quickly [32]. Three to five years are required to start seeing soil health benefits [33]. In northern Spain, which averages 1000 mm (40 in) of annual rain, the intensive rotational grazing of sheep resulted in higher forage production and increased carbon storage [34]. I traveled to parts of the world where intensive rotational grazing is really successful. These areas were in New Zealand, the United Kingdom, southern U.S., and Uruguay. Another issue that was reported is reduced livestock productivity [35]. This is most likely to occur when rotational grazing is first started. When the animals become accustomed to rotational grazing, this may be less likely to be an issue. Intensive rotational grazing is more likely to be successful on pastures with higher rainfall [36]. On more arid land, high, intensive, rotational mob grazing with frequent moves is more likely to reduce weight gain [37,38]. Research is needed to determine if low-stress cattle-handling methods would reduce this problem.

## 4. High-Plains United States Livestock Grazing

The high-plains area of the United States is in the midsection of the country and it includes Colorado, Kansas, Montana, North Dakota, and South Dakota. I have driven in this part of the country many times during both the summer and winter. The eastern parts of the high plains have sufficient rainfall for dryland farming and the more arid west plains are used for cattle ranching.

### 4.1. Grazing Cover Crops in Nebraska and Other High-Plains States

The research clearly shows that rotating cash crops, such as corn (maize) or soybeans, with a cover crop improves soil health [39]. To get the field ready for the next cash crop, the cover crop is often terminated with a herbicide, such as glyphosate [40]. From a sustainability point of view, this is a poor practice. There are many new research studies in Nebraska on grazing cover crops. Grazing cover crops is a win–win situation. It pays for both the cover crop [41,42] and provides food for people from the livestock. Grazing also greatly reduces the use of herbicides and chemical fertilizer. The cattle or other livestock are used to terminate the cover crop instead of using herbicides [40].

Farmers are often concerned that grazing a cover crop and having animals walking on the field could cause soil compaction and hurt the yields of their crops. A three-year study conducted in Nebraska showed that if stocking density is kept low, grazing a rye cover crop would have no effect on corn yield [43]. Other studies have also shown that grazing a cover crop had no effect on grain, corn, or soybean yields [44,45].

To obtain the maximum benefits, cattle are often grazed on both the cover crop and the stubble that is left over from the harvested corn (maize) crop [44]. On a recent drive through Nebraska, I observed many fields where cattle were grazing corn (maize) stubble. One study in Nebraska showed that an oats–rapeseed mixture cover crop was good for growing cattle before they are put into feedyards [46]. This is another win–win situation because growing young cattle on cover crops reduces the amount of grain that is fed to them. This is due to the animal being larger and heavier before it is placed in a feedlot.

### 4.2. Rotational Grazing on the Nebraska Sand Hills and Eastern Colorado

I have driven through the sand hills many times and there is no way that most of this land can be used for growing crops. It consists of steep sandy hills with small areas of flatland. On this land, Nebraska ranchers often used a four-pasture rotational grazing system. This had the added benefit of increasing beneficial dung beetles [47]. The plant diversity index was worse in the no grazing and continuously grazed pastures, compared to pastures that were rotated [47].

High-intensity mob grazing with high stocking rates works really well in areas with high rainfall [34]. In arid areas with lower rainfall, two long-term studies showed that it caused decreased cattle weight gain in Eastern Colorado and the Nebraska Sandhills [38,48]. In the Nebraska study [48], fencing methods and the use of small numbers of yearling steers in three different rotational pasture treatments may have influenced the results. When the steers were in the high-density mob condition, a portable electric fence was used to corral 36 steers and move them twice each day. In two other more lightly stocked rotational treatments, 8 to 9 yearlings were moved every ten days and permanent electric fencing was used. The differences in handling fencing and very small groups may have been one cause of the poor mob grazing results. When either a permanent electric fence is used, or a fence is moved gradually across a field, the cattle will always know where it is located. I observed that when small numbers of cattle were held in small electrically fenced pens, the cattle were more likely to accidentally contact the fence. On a regular commercial operation, the groups of cattle would be larger. This makes it easier for the cattle to avoid being shocked.

Another study conducted in Colorado compared very-high stocking-density rotational mob grazing with continuous grazing. The very-high stocking density was detrimental to weight gains [38]. The next step of the research was to experiment with lower stocking densities and longer periods between rotations. A rancher in Colorado reported that he had good cattle performance and improved plant heterogeneity after three years. He moved his cattle every three to seven days, depending on grass growth. For many people, mob grazing where animals have to be moved at least once a day is too difficult. It is my opinion that more research is needed on rotational systems where the animals are moved every few days.

## 5. Grazing the Eastern Wetter Parts of the U.S.

In other parts of the U.S. where it is warmer and wetter, a system of ten rotational grazing pastures worked well [49]. In a six year trial on a native, tall-grass prairie, the rotational system was compared to continuous grazing. Cattle movement to the next pasture was based on the animals eating 50% of the grass instead of a set number of days. In southern Missouri, I observed that ranchers are improving plant diversity and improving soil health with six to eleven paddocks. The cattle were moved when 50% of the grass was consumed. Moves during the growing season were every five to eleven days. Intensive rotational grazing with more frequent moves was beneficial in the warmer wetter parts of the U.S., such as Georgia and Mississippi [50,51].

In many parts of central and southern U.S., cattle graze fescue grass [52]. One of the problems with fescue is that most of this grass is infected with fungal endophyte, which increases the plant’s tolerance to environmental stresses [52]. To reduce toxicity to cattle, ranchers often seed clover in their fescue pastures [52]. Endophytic-infected fescue can impair the animal’s ability to thermal regulate [53]. Other methods to reduce fescue toxicity is breeding cattle to tolerate fescue [54]. Another study showed that the microbiota in the animal’s gastrointestinal tract had an effect on fescue toxicity [54]. Ranchers have had to learn how to adapt to endophytes infected fescue because fescue grass that lacks the endophyte is less tolerant of environmental stressors [55].

### Increasing Rancher Interest in Grazing

All over the U.S., there is increasing interest in using grazing to help sequester carbon and reduce the amount of grain fed to cattle [56,57]. There is also research that shows that targeted grazing can be used to improve rangeland [58]. When this method is used, livestock are brought in to manage a specific area by having animals eat plants that they want to eliminate. In some parts of the U.S., such as West Texas, the ground had to be converted to grazing because there was no longer sufficient groundwater to irrigate with center pivot sprinklers. While flying over this area in a commercial airliner, I observed that many of the center pivots were turned off. There were brown circles next to circles green with growth. In the more arid parts of the world, grazing is the only method for raising food on the land.

I emphasized that good managers could use cattle, sheep, and goats to improve the land. I traveled to all the areas of the U.S. where the land is grazed. During these travels, I observed both the best rotational grazing where the land was improved and the worst overgrazed fields where the land was degraded. I has also had the opportunity to observe land that was not grazed adjacent to arid land that was managed by good, conventional rotational grazing. The grazed land was in much better condition. The ungrazed pasture was full of dead grass surrounded by bare dirt. The grazed pasture had a much more even growth of grass. Ranchers are adopting rotational grazing that ranges from intense mob grazing to conventional pasture rotations. A recent survey showed that 60% of U.S. ranchers use some form of rotational grazing [59]. Recent discussions in 2022 with ranchers in southern Missouri indicated that about half the ranchers in the area used some form of rotational grazing.

## 6. South American Sustainable Grazing Systems

I had the opportunity to view well-managed grazing lands in Brazil, Argentina, Chile, and Uruguay. Uruguay is raising 90% of its beef on pasture [60]. In the more temperate parts of South America, there has been extensive research on integrating crops and livestock. Researchers have learned that soy monocultures are detrimental because key nutrients, such as potassium, are depleted [61]. In the temperate southern parts of Brazil, Holstein heifers are grazed on an integrated soy and ryegrass system. Many South American researchers are working to integrate livestock and crops [62,63,64]. Another truly sustainable system that utilizes livestock is Silvopasture systems. They are used in tropical and subtropical areas. They consist of both pasture and edible leaves or scrubs [65,66,67]. Silvopastures can be used to improve the soil and rehabilitate degraded pastures.

## 7. European Grazing Systems

In both western and eastern Europe, there is adequate rainfall and both rotational grazing and integrating crops with livestock has been successful [68,69,70,71]. In Germany, grazing of sheep and goats has been used to rehabilitate land that has been overgrazed by cattle [72]. These animals eat weedy, woody plants that cattle avoid. The research conducted in Hungary clearly observed that “one size fits all” strategies cannot be recommended [73]. They used a combination of sheep and cattle grazing. In Ireland, cattle are raised on pasture and housed in buildings in the winter because the fields are too muddy. The use of mixed grazing with sheep and cattle resulted in more diversity in both the vegetation and soil microbes [68]. The diversity of arthropods and birds was also improved with low-intensity grazing of mixed species [74]. I recently went on a tour of Southern Portugal. Cattle and sheep were being grazed among ancient olive trees. I believe that this practice was totally sustainable because the trees were not being damaged by the cattle and the ground had forage growing. Unfortunately, a growing export market for olive oil motivated producers to plant intensive irrigated monocultures of olive trees.

## 8. China and Northern Steppes Land Grazing

Many northern areas of China are more arid than Europe, the eastern U.S., or South America. China has large parts of the country that are grasslands [75]. Unfortunately, 90% became degraded [76]. This was due to overstocking the pastures, and not providing them with sufficient rest to recover. When pasture is severely damaged, excluding livestock increases soil organic carbon [77,78]. There is a point where exclusion provides no additional benefit and the introduction of well-managed rotational grazing improves organic matter [79]. Grazing does not need to be eliminated, but reducing “grazing pressure conserves diversity” [79]. Moderate grazing can improve species richness [80]. The key is just the right amount of moderate grazing [81,82].

An important pillar of sustainability is the local people should be able to have a viable livelihood. Traditional herding practices may mimic rotational grazing. A four-season nomadic herding system and a four-season rotational grazing were both less detrimental than “settle” continuous grazing [83]. Converting grassland to crops caused losses of nitrogen and soil organic carbon [79]. During my consulting business, I have been on extensive road trips. In many parts of China, I observed diverse agriculture with many different crops grown in strips on the same field. Few cattle or other grazing animals were visible. An area for possible future innovation would be incorporating grazing animals into these multiple crop systems.

## 9. Arid, Hot Australian Outback

A trip to the Australian Outback in 2018 made me think deeply about using livestock to produce food. I left Darwin and flew for two hours south in a small plane. I observed an arid vastness with no signs of houses, electrical wires, or roads. As I looked out over this vast land, I thought that people need to be able to raise food on this land. The plane flew almost directly over a single road and then, all of a sudden, a cattle station appeared. This station grazed cattle that were a mixture of European and Bos indicus breeds. Grazing cattle, sheep, or goats is the only way that food can be grown on this vast land. There is only enough water to water grazing livestock. Research is needed on the best way to manage this land. There were numerous studies done in other parts of the world. Outback land may be too arid and sparse to make intensive rotational grazing work. A nine-year study near Darwin showed that an intensive rotational system lowered weight gain [37]. In a study in the wetter, semi-arid parts of southeastern Australia, rotational grazing improved plant biodiversity [36].

## 10. Grazing in Africa

In South Africa, the integration of crops and grazing has similar results to other countries. Grazing livestock reduced the use of chemical herbicides [84]. In Africa, the vast semi-arid savannah pastures are inhabited by vast variety of grazing wild ungulates. The overstocking of pasture with cattle in these areas is detrimental to wild herbivores. However, lower stocking rates with cattle may represent a win–win for wild herbivore conversation and the individual performance of livestock [85]. Further research and innovation from local ranchers are required to develop sustainable grazing that can co-exist with wildlife.

## 11. Choosing the Right Livestock for Extensive Grazing for Both Productivity and Animal Welfare

I observed that the animal that performs best in an extensive grazing environment is not the same animal that performs well being fed grain in a feedlot. When I visited the hot Australian outback, I quickly learned that purebred British breeds could not survive there. Ranchers told me that they tried using Angus sires and they all died. This was a serious welfare problem. For heat tolerance, it is essential to have cattle with high percentages of Bos indicus breeding. These cattle can have good welfare if they eat sufficient forage to maintain body condition. U.S. ranchers learned that when cattle are selected solely for meat production, the result is often a cow that is too large and requires too many calories to be successful on extensive pasture. John Scasta, from the University of Wyoming, stated that the cattle industry encouraged ranchers to select large cattle that may not be suitable for “harsh rangeland environments [86]. Ranchers in Nebraska learned that moderate-sized, early maturing genetics cows are the best. Many ranchers told me that it was too expensive and difficult to feed larger cows hay in the wintertime. The original Angus and Hereford cattle from the United Kingdom were originally bred to fatten on grass. In the southern U.K., lush grass is plentiful. Producers who raise grassfed beef in the U.S. often use cattle that have not been bred for maximum meat production. I observed that the Hereford and Angus breeds have undergone extensive selection since the 1970s. Over the years, these breeds have been selected to fatten on grain. They were selectively bred to reduce the amount of external fat. Breeds, such as Murray Grey cattle, which have not been intensively selected for meat production, are popular in grass feeding programs [87]. In the U.S., in southern Missouri where it is colder and wetter, I observed that smaller, moderate-frame-sized cattle are being used at present.

## 12. Other Ways to Use Livestock to Improve Sustainability

### 12.1. Effects of Fire

Another use of grazing animals that needs further research is fire suppression. In Colorado, over 1000 homes were destroyed in a disastrous fire [88]. The fire started in a large, open, grassy area near the houses called Marshall Mesa Trailhead [89]. The grazing of this land near the houses might have prevented this fire. There are already some producers who are using grazing animals to reduce fire risk. Fire and planned burning is also needed to increase biodiversity on Savannah land [90]. I observed arid land in Arizona that has been ruined by invasive juniper bushes. This is not natural. University of Nebraska, Range Ecologist Dirac Twinwell is concerned that juniper bushes are taking over pastures [91].

### 12.2. Grazing under Solar Panels

Large amounts of land are now covered with solar panels. Innovative producers are now grazing sheep on the land that is underneath the panels [92,93]. This is another win–win practice. The sheep control weeds and mow the vegetation under the panels. Grazing sheep enable food production on land covered with solar panels. To graze cattle under solar panels requires the use of stronger supporting posts and the installation of the panels higher off the ground.

### 12.3. Combine the Best of Organic and Conventional Agriculture

To achieve the most sustainable system may require the use of many of the principles of organic agriculture [94]. A combination of organic and chemical fertilizer increased crop yield [94]. Both the scientific research and a discussion with farmers indicated that a total ban on synthetic fertilizer or crop chemicals may not be the best approach. One farmer told me: “It is a mistake to be too pure.” The use of rotational livestock grazing on cover crops makes it possible to greatly reduce chemical use. I predicts that the most sensible approach is the hybrid approach that combines the best principles of both organic and conventional agriculture. Merging a mostly organic approach with some use of industrial methods may work best [95,96]. Prices for conventional fertilizer have skyrocketed [97]. This is an additional motivator to integrate livestock and crops.

## 13. Animal Welfare Issues in Grazing Systems

When grazing is performed properly, the animals can have a high standard of welfare. Welfare scientists agree that preventing suffering is not sufficient. The animals should also have a “life worth living” and opportunities for positive experiences [98,99]. Good stockmanship is essential when livestock have to be continuously moved. Over the course of my long career, I have observed that cattle handling has greatly improved. Many people are now teaching low-stress methods. When handling is performed correctly, cattle become calmer when pastures are switched [100]. In very dry, extensive areas, such as the outback in Australia and desert grazing in Arizona, I observed that body condition must be closely monitored. In the wetter areas of the world, such as the green, lush pastures in Ireland, the cattle graze in the summer and they are kept in buildings in the winter because the fields are too muddy. The welfare of cattle in some of these barns may be compromised. I observed very filthy, dirty cattle arriving at slaughter plants in the U.K. and Ireland. The cattle originated from these barns. The successful use of Holstein heifers on a corn (maize) rye system in Brazil could serve as an example that could improve Holstein dairy cattle welfare. This would allow Holstein steers to stay on pasture and not enter a feedyard at a young age.

## 14. Local Grazing May Reduce Supply Chain Fragility

Local grazing operations may help provide food when there are supply chain disruptions. Everybody has now become more aware about the fragility of food supply chains. COVID-19 disrupted supply chains [101]. This has forced many people to become aware about how food enters their local supermarket. This has made more people become interested in local sources of food [102]. Before the COVID-19 pandemic, many people never thought about how both goods and food moved across their country or around the world. The public has now learned that big is fragile [103]. When COVID-19 sickened the workers in two large U.S. pork processing plants, over 300,000 pigs had to be destroyed on the farm [104]. This caused both a severe animal welfare issue and a huge waste of food.

The COVID-19 pandemic has motivated many U.S. cattle producers to start regional beef slaughter facilities [105,106,107]. These smaller plants will not be able to compete with the low-cost production of large processors. They will have to enter premium niche markets, such as grass-fed, family farm, organic, or locally raised, and charge more for their meat. Ranchers also need to receive good local advice on the use of both cover crops and pasture rotation. The most effective systems for both the pasture, rotation, and integration of crops and livestock needs to be adapted to local conditions. A system that works well in one part of the U.S. may not work in another part. One size does not fit all. I observed some early attempts at intense rotational grazing in arid Arizona in the 1970s that were not successful. The pastures were stocked too heavily and not given sufficient time to recover.

## 15. Conclusions

There are vast areas of land in the world that can only be used for grazing. Grazing makes it possible to produce food on land that cannot be cropped. Rotating grazing cattle or other livestock on a cover crop rotated with cash crops, such as corn (maize) or soy, will improve both soil health and reduce dependence on expensive, fragile fertilizer supply chains. The welfare of these animals is likely to be better compared to livestock housed indoors. There are many grazing alternatives that will be sustainable if the system is not overloaded. I observed that it was difficult for people to determine what was optimal production versus maximum production. Both the scientific research and practical applications indicate that both overgrazing and no grazing is detrimental to the land. There needs to be just the right amount of grazing. Well-managed grazing systems can be truly sustainable and improve soil health, help sequester carbon, and maintain plant biodiversity. The grazing animals are part of the cycle of life and the natural grass ecosystem. They are a natural part of the land.

## Data Availability

Not applicable.

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
