# Peer review of "Grazing Cattle, Sheep, and Goats Are Important Parts of a Sustainable Agricultural Future"

_animals, 2022, doi:10.3390/ani12162092_

Round 1
Reviewer 1 Report
Overgrazing is a global issue nowadays, and people are easy to jump to a quick solution-phasing out animal agriculture. However, animal agriculture is irreplaceable for current human society. This commentary provides a comprehensive review of the grazing systems in different regions, and summarizes the advantages of well-managed grazing systems. I want to appreciate the author for such a well-written and inspirational paper. I believe this commentary would make people think again about livestock grazing systems, and hope more people can read it.
I only have some suggestions about the format of this commentary.
Many two-spaces errors between sentences or in sentence should be corrected.
L30 1.Introduction
L40 bio-reactors
L140 [36].
L164 "benefits, cattle are"
L180 "mob grazing"
L183 "[38,48]. In the Nebraska study [48],"
L206 [49].
L224 [55,56].
L255 "[64,65,66]"
L256 pastures.
L303 [84].
L325 environments"
L333 "[88] in the U.S. In southern"
L344 "[91]."
L403 15.Conclusions
Author Response
RESPONSE TO REVIEWER 1
Thank you for your positive comments about my paper
Line 21 – Added the word “habitable” between “the” and “land”
Line 79 – Corrected reference number
Line 296 – Changed “the” to “and”
Line 402 – Deleted the dot in front of the Conclusions
I wish to thank all the reviewers for helping me to improve my paper.

Reviewer 2 Report
This article is a commentary of the author on the importance and proper use of grasslands for food production. The author has extensive first had knowledge of many livestock production systems around the world. My major concern with this article is the flow of thought. There is a lot of great information and examples, but the flow of thought is very broken (jumps from one idea to another without connection between thoughts). I suggest that a major re-write is necessary. I also suggest that the author include more of her personal insights and thoughts on the subject, and raise important questions with regard to use of grasslands for food production.
L47 - the large ruminants that were replaced by cattle also emitted methane such as bison. Hristov et al. 2012. Also, should discuss the new concept of GWP* such that methane has a 10 year cycle and because the amount of methane released is generally recaptured as CO2 in plant material, the amount of methane added is minimal
L49-53 - this idea should be a separate paragraph and be expanded upon. the author has so much knowledge to add to this idea
L64 - change to 'care deeply about the land'
L65 - what are these animal welfare benefits?
L78 - need a reference about the pollinating insects
L89 - change to 'sand hills for raising food is grazing livestock'
L91 - I wouldnt say surface of the earth unless you are counting ocean surface. I assume you mean 20% of land surface
L98-101 - this idea needs to be a separate paragraph and expanded further
L126 - change to 'stocked very densely'
L127-128 - these 2 sentences seem to contradict each other. First it talks about higher rainfall areas then says this is especially true of more arid areas.
L142 - change to 'Research is needed'
L174-179 - the flow of thought in this paragraph needs improvement
L180 - change to 'mob grazing'
L204 - did you mean eastern wetter parts of the US?
L205 - change to ' a system of ten pastures in rotational grazing system worked well'
L216 - change to 'tolerance to environmental stresses [52] but results in lower heat and cold stress tolerance of cattle [X]'
L221 - I wouldnt use the term weaker. I would say less tolerant of environmental stressors and less persistent
L239 - this is a description of the ungrazed pasture. a comparable description of the grazed pasture should be included
L269 - what does totally sustainable mean? How would you define it? How would you measure it to know if a system was sustainable or not? I am not sure you can say it was totally sustainable
L296 - change to 'a mixture of European and Bos indicus breeds'
L333 - Murray Grey are popular for what? grass-finished beef systems?
L383 - this section seems to be an after thought. It does not really fit with the main theme of using grasslands and livestock grazing for food production
Author Response
RESPONSE TO REVIEWER 2
I agree with Reviewer 2’s recommendation that I need to improve the flow of thought in my paper. I have completely rewritten the first part of my paper to improve it. Per Reviewer 2’s suggestion, I have added additional description about my personal experiences during travel to grazing lands located in the U.S. and in other parts of the world. I wish to thank Reviewer 2 for helping me to improve my writing.
Line 47 – Added the Hristov et al. 2012 reference on methane emissions from bison. It is in the rewritten part of the paper.
Line 49-52 – Added my own observations about consciousness in animals.
Line 64 – The section has been rewritten. On a different line – changed “to care deeply about the land.”
Line 65 – On a different line in the rewritten first part of the paper, the welfare benefits are stated. Added a sentences – On a well-managed grazing operation cattle and sheep welfare would be better compared to either indoor confinement or a muddy feedlot.
Line 78 – Added a reference on pollinators by Lee-Mader et al., 2014.
Line 89 – Removed the phrase “The best use of the” and replaced with the” Sandhills is being used for grazing livestock.”
Line 91 – Rewrote it to make it clear that 20% refers to habitable land on earth.
Line 98-101 – Removed this section because it interrupted the flow of thought.
Line 120 – Changed “tightly” to “densely”
Line 127-128 – Corrected the contradiction changed “arid” to “wetter”
Line 142 – Changed wording to “research is needed”
Line 174-179 – Changed wording to improve flow
Line 180 – Corrected typo and changed “brazing” to “grazing”
Line 204 – Corrected subtitle and added the word “parts”
Line 205 – Correct wording on rotational grazing
Line 216 – Added a reference – Mote et al. 2020 on impaired thermal regulation
Line 221 – Changed “weaker” to “less tolerant of environmental stressors”
Line 239 – Added a description of grazed pasture
Line 269 – Added an explanation explaining why I believe the grazing among ancient olive trees is sustainable.
Line 296 – Changed to a mixture of European and Bos indicus breeds
Line 333 – Explained why Murray Grey cattle are popular
Line 383 – Added that local grazing operations may help provide food when there are supply chain disruptions. This provides a link to the rest of the paper.
I wish to thank all the reviewers for helping me to improve my paper.

Reviewer 3 Report
It is a privilege for me to be able to read this paper before my colleagues.
In these turbulent times where dogmatism, frequently in the mouth of neophytes, invades us, the reflection of great researchers with extensive experience is needed, as is the case of Dr. Grandin.
All the text seems perfect and necessary to me, I do not have the slightest objection to point out.
Just three typos:
Line 79: I think we should change the reference number 20 to 21
Line 296: I think we should change the to and
Line 403: Delete dot in front of Conclusions
Author Response
RESPONSE TO REVIEWER 3
I really appreciated your comment that there is a need for well-managed grazing systems.
Thanks for correcting the formatting errors. I have gone through the paper and corrected them. The entire reference list got put in italics by accident. That has been corrected.
I wish to thank all the reviewers for helping me to improve my paper.
